



# Aeroelastic Tailoring of Wind Turbine Rotors Using High-Fidelity Multidisciplinary Design Optimization

Marco Mangano [1], Sicheng He [1], Yingqian Liao [1], Denis-Gabriel Caprace [2], Andrew Ning[2,3], and Joaquim R. R. A. Martins[1]

[1]University of Michigan, Ann Arbor, Michigan, United States
[2]Brigham Young University, Provo, Utah, United States
[3]National Renewable Energy Laboratory, Golden, Colorado, United States

**Correspondence:** Marco Mangano (mmangano@umich.edu)

**Abstract.** Reducing the cost of energy from wind power is a critical step towards decarbonizing the electric grid and mitigating climate change effects. Computational models are crucial in understanding the complex, multiphysics interactions of modern, highly flexible wind turbine rotors. When coupled with numerical optimization, such models provide an efficient way to explore the design space. We propose a high-fidelity aerostructural framework that couples computational fluid dynamics with

computational structural mechanics to analyze and optimize wind turbines. The framework uses a gradient-based optimization strategy with gradients efficiently computed using a coupled-adjoint approach. We optimize a benchmark utility-scale wind turbine rotor and explore its trade-offs between steady-state aerodynamic efficiency and structural weight. The optimizations account for a representative below-rated operating condition and use more than 100 structural and geometric design variables. The monolithic approach we propose is compared with a loosely-coupled optimization strategy used for the structural sizing

of the baseline rotor layout. We then discuss specific optimized blade features and a broader design trade space exploration between extracted torque and rotor mass. Optimized layouts increase the torque by up to 14% and reduce the mass by up to 9% or reduce the mass by up to 27% with the same torque output compared to the baseline blade. The results demonstrate the benefits of optimizing a tightly-coupled aerostructural model and reveal additional insights that high-fidelity analysis provide to complement conventional design approaches.

**Nomenclature**

**AEP**   annual energy production

**BEMT**   blade element momentum theory

**CFD**   computational fluid dynamics





**CSM**  computational structural mechanics

**FFD**  free-form deformation

**IEA**  International energy agency

**KS**  Kreisselmeier–Steinhauser

**LCOE**  levelized cost of energy

**MACH**  MDO of aircraft with high-fidelity

**MDA**  multidisciplinary analysis


**MDO**  multidisciplinary design optimization

**RANS**  Reynolds-averaged Navier–Stokes

**TACS**  toolkit for analysis of composite structures

**TSR**  tip speed ratio

**Tw**  structural and twist variables


**TwSc**  structural, twist, and scaled blade section variables

**TwChTk**  structural, twist, chord, and relative thickness variables

**XDSM**  extended design structure matrix



# 1 Introduction

Climate scientists overwhelmingly agree that anthropogenic emissions are altering the Earth's atmosphere temperature and composition. Experts committees such as the Intergovernmental panel on climate change (IPCC) (Masson-Delmotte et al., 2018) predict that such alteration will negatively impact natural resources availability, industrial-scale food production, and ultimately the well-being of current and future generations. The International energy agency (IEA) (Cozzi et al., 2021) has proposed net-zero emission targets to mitigate global warming and reverse the effects of the last two centuries of industrial emissions. A rapid and systemic transition to affordable carbon-free energy sources is necessary to limit the global temperature increase to 2.0°C by the end of the century. According to IEA, almost 70% of the global energy needs in 2050 should be harvested from solar and wind resources.

Onshore wind energy is a mature technology on an industrial scale, and there is a growing interest in developing more extensive and efficient offshore installations. Like other carbon-free energy sources, resource intermittency and installation-site constraints currently limit the amount of energy captured. Industry trends, driven by the need to mitigate these issues, suggest that wind turbines and their rotors will keep growing in size in the next decades (Veers et al., 2019, 2022). Large-diameter rotors with low solidity increase the power output and exhibit better aerodynamic efficiency (Hau, 2006). Turbines with large rotors can be deployed onshore and offshore. Onshore, these turbines extend the operating range to lower wind speeds. Offshore deployment circumvents transportation constraints and exploits the untapped potential of wind away from the coastline (Musial et al., 2016).

Numerical simulations are essential to model the multiphysics interactions that drive the performance of wind turbines and accelerate their development (Jonkman, 2009). However, the traditional sequential design of aerodynamic shape and internal structure does not take full advantage of the interaction between fluid and structural dynamics (Bottasso et al., 2016; Barrett and Ning, 2018; Martins and Lambe, 2013). Monolithic multidisciplinary design optimization (MDO) approaches exploit trade-offs in tightly-coupled aerostructural models to ultimately drive down the cost-of-energy of wind turbines (Garcia-Sanz, 2020). However, aerostructural models are challenging to implement within gradient-based MDO because they require accurate and efficient coupled derivative computation to guide the optimizer toward the optimal design.

In this work, we propose the first application of a high-fidelity aerostructural optimization tool for wind turbine design. We build on our previously developed software, MDO of aircraft with high-fidelity (MACH) (Kenway et al., 2014; Kenway and Martins, 2014), and previous work on computational fluid dynamics (CFD)-based approach aerodynamic shape optimization (Dhert et al., 2017; Madsen et al., 2019).

Madsen et al. (2019) used a Reynolds-averaged Navier–Stokes (RANS)-based aerodynamic shape optimization framework to optimize a turbine rotor's aerodynamic performance by varying a large set of design variables that parametrized the blade planform and cross-sectional shape. This paper extends the analysis and optimization to a tightly-coupled, steady-state aerostructural model. We optimize the rotor blades by considering the aerostructural performance at below-rated operating conditions. The combination of CFD and computational structural mechanics (CSM) expands the optimization design space and improves



the fidelity compared to blade element momentum theory (BEMT) and beam-theory-based design tools. We discuss how our work relates to the broader literature in Sec 1.1.

High-fidelity optimization can become intractable when evaluating many design points, regardless of the framework and implementation efficiency. In this paper, we demonstrate the developed capability for a single inflow condition for demonstration purposes and to limit the computational cost. We extend the framework to handle many more conditions in another parallel effort (Caprace et al., 2022), where we combine high-fidelity and conventional BEMT in the optimization to consider fatigue and extreme loads in the blade sizing. Combining the high-fidelity approach with conventional, full-system analysis codes can maximize the performance and minimize the design costs of the next generation of wind turbines, considering steady-state and life-cycle performance and operations.

The present work expands and supersedes the conference proceedings of Mangano et al. (2022) by using updated tools and models to increase the complexity of the optimization studies and extend the design space exploration scope. The following section reviews the state-of-the-art for wind turbine design optimization and highlights our contributions to large wind turbine rotor designs.

## 1.1 State-of-the-art for Wind Turbine Design Optimization

Several tools have been developed to design wind turbines over the last decade. State-of-the-art software, such as Open-FAST (NREL, 2018), Cp-Max (Bottasso et al., 2016), HAWTopt2 (Zahle et al., 2016), ATOM (Scott et al., 2019), SHARPy (del Carre et al., 2019), Qblade (Marten et al., 2013), and MoWit (Leimeister et al., 2021), simulate the full turbine system, including aerostructural rotor and tower models, the power generator, and fixed-bottom foundations or floating platforms. Linear and nonlinear controllers are also included for generator and pitch controller development, adding more realism to the modeling of the turbine operating behavior. Such a holistic approach is necessary to capture the system's multibody dynamics and the unsteady aerostructural phenomena, which drive the performance and sizing of a turbine over its life cycle. We refer to these tools as *conventional* to distinguish them from the framework presented in this paper.

These conventional tools combine models of variable fidelity to analyze different system components. OpenFAST (NREL, 2018) has set the standard for turbine analyses, combining a modal and multibody-dynamics formulation that includes all the main components. OpenFAST models the structure using the BeamDyn and ElastoDyn modules, which use beam models to capture the turbine dynamic response. However, these modules are limited in terms of degrees of freedom (ElastoDyn) and accuracy of local stress values (ElastoDyn and BeamDyn). OpenFAST's aerodynamic predictions through BEMT-based AeroDyn agree well with experiments for conventional designs. However, BEMT assumptions become increasingly poor for larger blades with large deflections and more significant 3D effects.

The blade-resolved rotor model used by Madsen et al. (2019) and in the present work also neglects the hub geometry. However, the CFD analysis captures the spanwise and tip flow phenomena with a more significant impact on in-plane and off-plane loads. Horcas et al. (2022) showed how a tool based on the same physical models of OpenFAST, HAWC2, overpredicts the loads on a 10 MW turbine at below-rated wind speeds and underestimates the benefits of curved wing tips, even for a steady power curve. Moreover, tools such as OpenFAST are limited when used with numerical optimization because they do



not compute accurate gradients. Furthermore, these tools use precomputed wing section drag polars and thus cannot optimize
the airfoil shapes.

Several efforts combined high- and low-fidelity tools to balance accuracy and computational cost. Ramos-García et al.
(2021) coupled finite-element structural models with a hybrid lifting-line-vortex method in a multibody aero-hydro-servo-
elastic framework, showing good agreement with Heinz et al. (2016). This effort addressed some limitations of BEMT at
a lower cost than blade-resolved solvers. However, the application of such tools is primarily limited to analyses because
computing derivatives remains challenging.

Other recent efforts focused on a narrower but higher-fidelity approach to capture the complex fluid-structure interaction of
the turbine blades (Lee et al., 2017). Wainwright et al. (2021) combined a commercial CFD software and a finite-element modal
solver with GPU acceleration to study the aerostructural behavior of a rotor in the wake of an upstream turbine. They used a
strong coupling strategy to resolve time-accurate analyses, using a radial basis function to pass load-transfer information be-
tween the solvers. Cheng et al. (2019) validated a model that couples an actuator line model, an aerodynamic RANS solver, and
a hydrodynamic CFD tool in OpenFOAM to study the behavior of a floating wind turbine. While BEMT and beam models have
limited accuracy for complex flow conditions, high-fidelity codes are limited by their computational cost and implementation
complexity. Hence, despite the promising results, none of these high-fidelity models have been used for optimization.

There has been some work on applying numerical analysis tools to wind turbine optimization problems. Gray et al. (2014)
and Ning and Petch (2016) showed how analytical gradients can be efficiently computed to handle optimization problems
beyond rotor design and include a more extensive set of constraints. The problem formulation has been extended to include
fatigue considerations in the structural sizing (Ingersoll and Ning, 2020). The design space exploration strategy developed by
Bortolotti et al. (2020) efficiently sized the rotor at a conceptual design stage, informing the rotor development. However, they
could not exploit the trades between the aerodynamic and structural design, yielding suboptimal designs from an aerostruc-
tural standpoint. Bottasso et al. (2012) proposed a bi-level design approach in a loosely-coupled nested optimization using
a comprehensive turbine model with a broad range of design constraints. They found the design maximizing annual energy
production (AEP) over the mass by interpolating a family of designs generated via a low-fidelity optimization.

A more compact sequential approach using the same tool was presented by Bortolotti et al. (2016), while another work from
the same research group extended the model to a monolithic approach that includes a more accurate FEM analysis to evaluate
the structural properties of selected blade cross-sections (Bottasso et al., 2016). The latter method eliminated any simplifying
load-displacement transfer assumption between the aerodynamic and structural models, but the algorithm was described as
"more complicated and in general less robust" than the sequential counterparts.

To avoid the implementation effort of a tightly coupled aerostructural approach, the sequential approaches use a fixed or par-
tially updated set of optimized aerodynamic loads between structural optimization sub-iterations. The iterative algorithm drives
the aerodynamic and structural models to numerical convergence, but the resulting configuration does not achieve the perfor-
mance that would be possible with a tightly coupled approach. Scott et al. (2020) used a similar approach for the sequential
optimization of a large turbine using a linearized aerostructural model to speed up aero-servo-elastic simulations.



Zahle et al. (2016) and McWilliam et al. (2018) showed the potential of aeroelastic tailoring and passive load alleviation in blade design when utilizing composite material anisotropy. Although they used conventional low-fidelity models and finite-difference gradients, which ultimately limit the optimization accuracy and robustness, their results were promising. Heinz et al. (2016) investigated the coupling of a CFD and an aeroelastic solver, showing good agreement with conventional approaches despite a non-conservative force transfer scheme and a loosely-coupled aerostructural solution.

Commercial CFD codes and reduced-order structural models have also been used to analyze the blade-tower interactions and quantify acceptable tip clearances and efficiency losses (Horcas et al., 2016). Even these latter efforts did not perform CFD-based multidisciplinary optimization using a monolithic architecture, which is the main contribution of this paper.

## 1.2 Contributions and Outline

This paper presents the first aerostructural optimization study for a wind turbine rotor using a coupled CFD-CSM solver. We leverage the high fidelity of our aerostructural analysis to capture steady-state aeroelastic interactions over the rotor of the DTU 10 MW configuration. The optimization is computationally tractable thanks to the efficient computation of gradients via a coupled-adjoint method with a gradient-based optimizer. The objective function is a metric combining rotor torque and mass. This metric can be adjusted to prioritize aerodynamic efficiency or mass reduction. The optimization concurrently varies more than one hundred aerodynamic and structural design variables.

This paper has several original contributions. As discussed in Sec 1.1, only a handful of previous efforts solve the same optimization problem with a gradient-based approach and a monolithic or semi-monolithic architecture. None of these efforts use a 3D coupled high-fidelity model, which has fewer simplifying assumptions than low-order models and provides greater design freedom. Our CFD-based aerodynamic analysis does not require pre-computed lift-drag polars, enabling the parametrization of the blade outer mold line and its airfoil sections. Similarly, the 3D structural model enables a component-based parametrization that is more detailed than an equivalent beam model and avoids sectional stiffness property condensation. Thus, this work is the first to perform concurrent aerodynamic and structural design optimization of a blade-resolved turbine rotor model.

On the multidisciplinary coupling side, the geometry of the aerodynamic and structural models is consistently updated, and the force-displacement transfer is conservative, ensuring physical consistency between the analyses. The tight coupling of these two models ensures the physical and numerical consistency of steady-state rotor aeroelastic response. The adjoint method for computing derivatives ensures scalability with increasing number of design variables (Martins and Ning, 2021, Sec. 6.7). Modeling assumptions and limitations still exist in the developed framework, but we can explore rotor design in a way that conventional optimization approaches cannot.

We describe our framework in more detail in Sec. 2 and the baseline rotor we optimized in Sec. 3. The problem formulation is discussed in Sec. 4, where we define the objective, constraints, and design variables. Section 5.1 presents the results for the mass minimization studies, highlighting the pros and cons of tightly-coupled optimizations. In-depth aerostructural optimization studies using structural and geometric design variables are discussed in Sec. 5.2. Finally, Sec. 5.3 presents a design space exploration using different objective function formulations. The key takeaways of our approach and results are summarized in Section 6.





## 2 Methodology

Gradient-based optimization is the only viable strategy to solve high-dimensional and high-fidelity optimization problems (Martins and Ning, 2021, Sec. 1.5). The cost of the adjoint approach for coupled system derivative computation is independent of
the number of geometric and structural design variables, making the approach advantageous for optimization problems where the number of design variables is greater than the number of functions of interest (Martins and Ning, 2021, Sec. 6.7).

We use our MDO framework, MACH, to perform aerostructural optimization studies using high-fidelity CFD and CSM analysis tools Kenway et al. (2014). This framework uses a coupled-adjoint approach for gradient computation and has been extensively used for wing (Kenway and Martins, 2014; Liem et al., 2015; Burdette and Martins, 2018; Brooks et al., 2018, 2019;
Bons et al., 2022; Brooks et al., 2020; Bons and Martins, 2020) and hydrofoil design (Garg et al., 2017, 2019; Liao et al., 2021, 2022). This paper applies the same methodology to a high-fidelity wind turbine aerostructural model to demonstrate the potential performance and mass reduction benefits of integrating this approach in rotor blade design.

As shown in the extended design structure matrix (XDSM) diagram (Lambe and Martins, 2012) in Figure 1, MACH is composed of a set of tightly-integrated sub-modules that enable geometry parametrization and deformation, coupled aerostruc-
tural analysis, and efficient derivative computation for gradient-based optimization. MACH-Aero [1] has been applied to a broad range of aerodynamic shape optimization studies, including unconventional aircraft (Lyu and Martins, 2014; Secco and Martins, 2019; Yildirim et al., 2021), hydrofoils (Liao et al., 2020), and wind turbines (Dhert et al., 2017; Madsen et al., 2019).

The geometry manipulation module, pyGeo [2], is based on the free-form deformation (FFD) approach (Sederberg and Parry, 1986) as implemented by Kenway et al. (2010). Geometric design variables are defined through the displacement of a set
of control points over the FFD volume, as further discussed in Sec. 4.2. The aerodynamic surface and structural meshes are embedded in the same control volume, so the control point displacement consistently updates both models. The aerodynamic surface node displacements are propagated to the volume mesh through IDwarp [3], which uses an interpolation algorithm based on inverse distance weighting method (Luke et al., 2012). This avoids the need to regenerate the CFD mesh at each iteration, increasing the optimization robustness and reducing the computational cost.

For aerodynamic analyses, we use ADflow [4] (Mader et al., 2020), a second-order, finite-volume compressible solver developed at the MDO Lab. We use the Spalart–Allmaras (Spalart and Allmaras, 1992; Lyu et al., 2013) turbulence model and leverage on a low-speed preconditioner and an approximate Newton–Krylov method (Yildirim et al., 2019) to minimize accuracy and computational cost penalties at low inflow speed (Madsen et al., 2019; Chauhan and Martins, 2021). We do not model transition in our analyses, although the capability is under development Shi et al. (2020, 2021). ADflow includes an efficient
formulation to compute flow derivatives via the adjoint method (Kenway et al., 2019), which makes it ideal for gradient-based optimization.

---

[1] https://github.com/mdolab/MACH-Aero

[2] https://github.com/mdolab/pygeo

[3] https://github.com/mdolab/idwarp

[4] https://github.com/mdolab/adflow





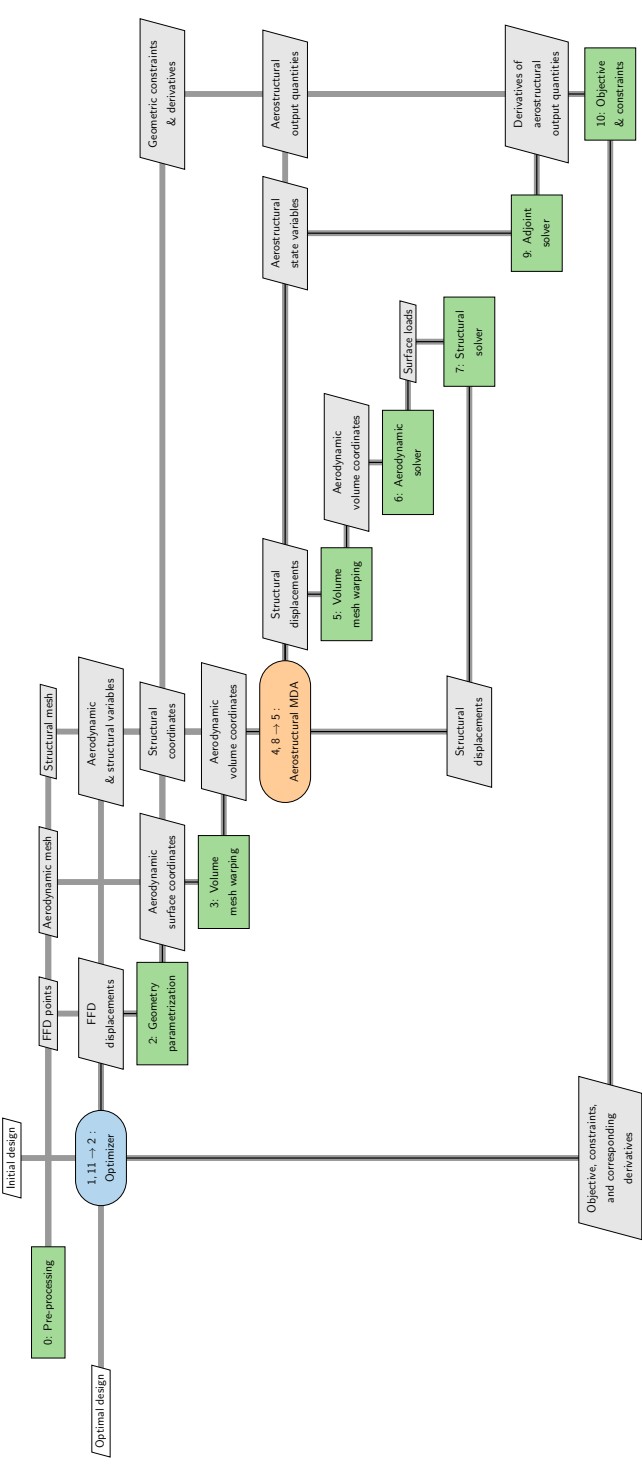

**Figure 1.** The MACH optimization framework combines several modules for model manipulation and analysis.



The structural solver used in this work is the toolkit for analysis of composite structures (TACS) (Kennedy and Martins, 2014). This solver is scalable, robust, and computes accurate machine-precision derivatives using the adjoint approach. This tool has been used in several previous studies and includes nonlinear (Gray and Martins, 2021) and stability (Jonsson et al., 2019a) analysis capabilities. In this work, we perform linear structural analyses with a preconditioner based on the direct factorization of the global Schur complement matrix and a modified GCROT linear solver (Hicken and Zingg, 2010). The structural stress constraint we use in this work is based on the von Mises criterion (Kennedy, 2016). The assumptions on our structural model are discussed in Section 4. Our previous paper also included tip displacement constraints (Mangano et al., 2022), which will be the object of future studies.

We use the Kreisselmeier–Steinhauser (KS) aggregation function (Kreisselmeier and Steinhauser, 1979), which can be used to reduce the cost of the adjoint gradient computations (Martins and Ning, 2021, Sec. 5.7). The KS function yields a smooth approximation of the maximum value of constraints over the complete set of element-wise local function values (Lambe et al., 2017; Kennedy and Hicken, 2015).

A nonlinear block Gauss–Seidel (NLBGS) solver converges the coupled aerostructural analysis [Sec.13.2.4](Martins and Ning, 2021). The linear coupled-adjoint system is solved via the coupled Krylov (CK) subspace method, as implemented by Kenway et al. (2014). Finally, pyOptSparse [5] (Wu et al., 2020) is used to pass the function values and sparse Jacobian from MACH to the optimizer (SNOPT (Gill et al., 2005)).

## 3   Baseline Turbine Rotor Configuration

In this study, we use the DTU 10 MW benchmark rotor configuration as the baseline design for our optimizations, building on the work of Madsen et al. (2019). This is a three-blade rotor with 178.3 m diameter and tailored chord, airfoil, and twist distributions. The outer mold line and structural layup are defined by Bak et al. (2013).

The blades are blended at the root to limit the meshing effort and prevent CFD solver convergence issues (Dhert et al., 2017). This neglects the hub effects on the aerodynamic performance. Although the aerodynamic blade-nacelle interactions are a relevant design factor, they have a limited impact on the extracted torque for large turbines. The influence of these interactions on the blade structural sizing is negligible. Tower-blade interactions are also ignored.

Our simulations use the same aerodynamic meshes as those of Madsen et al. (2019). A family of four meshes was defined, starting from the most refined L0 (14 155 776 cells) to L3 (27 648 cells) in increasing coarsening levels. Between the L0 and L1 meshes, we remove every other node in the three dimensions, reducing the number of cells by a factor of 8. This is repeated for L2 and L3 meshes. ADflow is particularly sensitive to mesh resolution in the incompressible flow regime (Madsen et al., 2019). In this paper, we use the L1 mesh for optimization to balance accuracy and computational cost. This mesh has 1 769 472 volume cells, and a single aerodynamic analysis converges in less than 30 CPU-hours.

---

[5]https://github.com/mdolab/pyoptsparse





## 3.1 Structural Model

The mesh follows the outer mold line contour and includes an internal main shear web and a reinforcement spar in the trailing edge area. CQUAD hexahedral linear shell elements are used to model the structural components. The mesh has $208\,464$

elements. Three identical blades are fixed at the root, located at $3\,\mathrm{m}$ from the rotation axis. The sizing process for the baseline rotor is outlined in Sec. 5.1

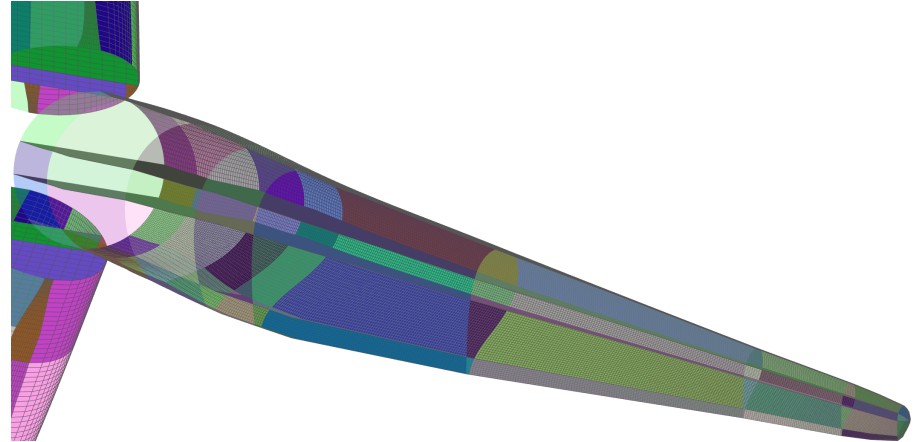

**Figure 2.** Color patches represent panel thickness design variables in the rotor blade internal structure.

We use isotropic shell elements made of 2024 aluminum alloy. This structural model is relatively simple because this paper's primary objective is to demonstrate our optimization framework's capabilities. In future work, we plan to perform coupled optimization studies with a composite model and additional design variables, such as fiber angle and ply fractional layup, to

leverage structural anisotropy and tailored bend-twist coupling properties (Brooks et al., 2019; Liao et al., 2021).

Figure 2 shows the structural model. The parametrization of the different structural panels independently sized by the optimizer is also simplified at this stage. The current layout is a compromise between design freedom and complexity. The panel discretization follows the main geometry breaks of the outer mold line. Our previous work (Mangano et al., 2022) compared different parametrization approaches, and future developments of the structural model will analyze the effects of structural

design variables distribution in more detail. The structural analysis with fixed loads is faster than the aerodynamic analysis by more than four orders of magnitude on average, considering the L1 aerodynamic mesh we use in this study.

## 4 Problem Formulation

The multidisciplinary optimization problem considers the rotor's mass and efficiency in harvesting power from the wind. These two factors, extended to the complete system and ultimately up to power plants, drive the cost-of-energy minimiza-

tion (Garcia-Sanz, 2020). A set of geometric and structural variables tailor the blade design and its steady-state aeroelastic





response, exploiting the trade-offs between aerodynamic performance and weight reduction while satisfying nonlinear structural and performance constraints.

The general constrained optimization problem formulation is as follows:

minimize $\mathbf{f}(\mathbf{x})$

by varying $\mathbf{x}_{\min} \leqslant \mathbf{x} \leqslant \mathbf{x}_{\max}$

subject to $\mathbf{g}(\mathbf{x}) \leq 0$

where $\mathbf{f}(\mathbf{x})$ is a designer-defined objective function that quantifies the merit of the design (Sec. 4.1), $\mathbf{x}$ the complete set of design variables (Sec. 4.2), and $\mathbf{g}(\mathbf{x})$ a set of linear and nonlinear constraints (Sec. 4.3). The upper and lower bounds of the design variables $\mathbf{x}_{\max}$ and $\mathbf{x}_{\min}$ are handled separately from other analysis-dependent constraints $\mathbf{g}(\mathbf{x})$. We expand on these

functions and problem variables in the following sections.

## 4.1   Optimization Objective

The cost of energy is driven by the turbine power output and the sum of investment and operating costs. The levelized cost of energy (LCOE) is a popular design objective because it takes these factors into account over the entire lifetime of a power plant (Dykes et al., 2014; Ashuri et al., 2014; Ning et al., 2014; Myhr et al., 2014). Recent research efforts have included produc-

tion intermittency to more accurately estimate the plant economic return considering the grid integration challenges (Simpson et al., 2020).

The optimization studies in this paper are limited to a single turbine rotor at a single inflow condition, so LCOE and comparable metrics cannot be directly computed for our model. We start from the metric proposed by Garcia-Sanz (2020) and simplify for an individual turbine with fixed inflow velocity and swept area, as explained more in detail by Mangano et al.

(2022). This results in using torque as a proxy for aerodynamic performance and rotor mass as a proxy for cost. The cost estimate assumes that the manufacturing process does not substantially change for the optimized layouts. The tower structure is a significant cost driver for turbines, but it is not directly considered in this work. Nevertheless, because our optimization starts from an existing benchmark design, we enforce a thrust constraint (Sec. 4.3) in our formulation to ensure that the loads from the rotor to the tower remain within a threshold relative to the baseline.

We use two approaches to explore the design space. The first approach combines the torque and mass metrics are into a single objective function as a weighted sum,

$$f(\mathbf{x}) = -\omega \frac{Q_x}{Q_{x_i}} + (1 - \omega) \frac{M}{M_i}, \tag{1}$$

where $Q_x$ and $M$ are the torque and mass, respectively, and $0 \leq w \leq 1$ is an arbitrary coefficient. Both metrics are normalized with respect to their baseline values $Q_{x_i}$ and $M_i$. The negative sign in the torque component is necessary because the optimizer

minimizes the function $f(\mathbf{x})$. Solving the optimization problem defined by Equation (1) for values of $\omega$ ranging between 0.7 and 0.5 results in a Pareto front defined by the torque and mass objectives. In these studies the optimizer prioritizes the torque output without penalizing the rotor mass or violating the thrust constraint.





The second approach uses the epsilon-constraint method to explore the Pareto front (Martins and Ning, 2021, Sec.9.3.2). Using this method, we perform mass minimizations ($f(\mathbf{x}) \equiv M/M_i$) with a constraint on the torque. We vary the torque

constraint value to add points in the Pareto front more accurately.

The first approach explores designs with more significant torque increases. The second approach populates the Pareto front closer to the baseline design. We discuss which approach is more appropriate for a given study in Sec. 5.

## 4.2 Design Variables

The FFD approach (Sec 2) applied to the blade-resolved layout of the aerodynamic and structural meshes enables a flexible

and effective parametrization of geometry components whose shape would otherwise be approximated or kept fixed through the optimization. Conventional approaches for turbine design are either limited in the number of high-level parameters directly used within an optimization or require additional external iterations to update blade section drag polars and stiffness properties between optimization loops. The high-fidelity approach enables the optimizer to alter detailed layout features, such as airfoil shape and individual structural panel thicknesses, within a monolithic optimization run. This reduces human intervention in

the loop and avoids the discontinuities in the design space introduced by conventional multi-level strategies. In addition to the increased control on local variables for design and manufacturing considerations, using a representative 3D model gives a more accurate system mass estimation using an element-by-element calculation.

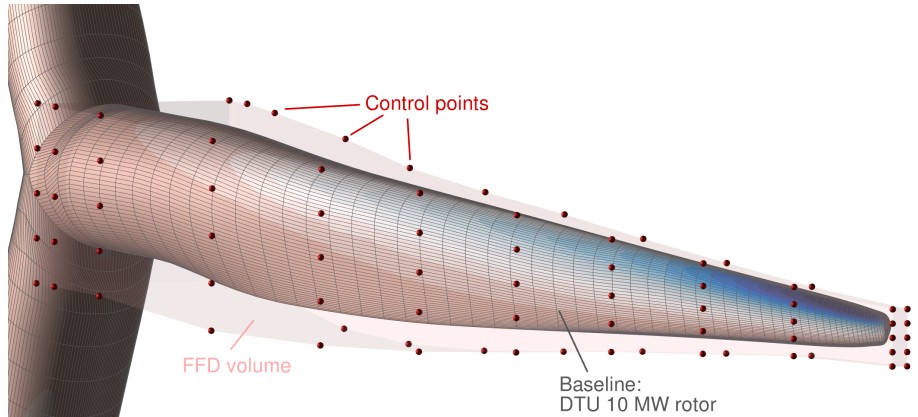

**Figure 3.** The rotor blades (with highlighted pressure contour here) are embedded in the FFD volume, and the shape is manipulated through control points displacement.

A snapshot of the aerodynamic model embedded in the FFD control box is illustrated in Figure 3. We group the control points of the pre-processed FFD grid and parametrize their displacement through user-defined functions passed to pyGeo. In

this way, we model planform design variables and local airfoil shape deformations using the same geometry object. The pitch and twist variables are defined as rotations of the FFD grid control sections around a user-defined axis. Chord and thickness changes are obtained by scaling the control sections along one of their axes. This work uses either the same factor to scale





chord and thickness (preserving the airfoil shape) or decoupled coefficients to give the optimizer more design freedom. The former approach mimics the parametrization capabilities of conventional design tools. With the latter approach, the independent

scaling of chord and thickness changes the airfoil section shape in a way that goes beyond the capabilities of these lower-fidelity tools.

The parametrization scheme can model additional geometric design variables not currently included in our optimizations. Pre-cone and pre-bend can be obtained by displacing airfoil sections off-plane, while airfoil shape can be changed by displacing single control points (Madsen et al., 2019).

We split the mesh into nine spanwise sections to parametrize the structural model. We further split the upper and lower skins into four chordwise sections. Each of these structural panels is assigned an independent design variable. This results in 117 structural thickness design variables, as shown in Figure 2. Future work will focus on more detailed structural parametrization and anisotropic material properties, enabling additional design variables, such as individual composite plies angle and thickness.

## 4.3 Constraints

The structural model addresses structural feasibility considerations and avoids the need for geometrical thickness and root bending moment constraints that are usually enforced in aerodynamic-only shape optimization studies (Madsen et al., 2019; Dhert et al., 2017). TACS is used for structural sizing, taking into account local element-wise stresses.

To implement stress constraints efficiently and effectively for gradient-based optimization, we use the KS constraint aggre-
gation technique mentioned in Sec. 2. This is useful for two reasons. Firstly, constraining the mechanical stress shell-by-shell would increase the number of functions of interest for the optimization by up to six orders of magnitude, negating the benefits of the adjoint method for derivative computation. Secondly, a function that only considers the overall maximum stress over the blade would be non-smooth because the most critically stressed element can vary between successive optimization iterations, hindering optimization convergence.

We define three aggregated KS stress constraints for this problem: the blade upper skin, lower skin, and spars, respectively. The inflow condition we are considering is not critical for structural sizing, so the optimized rotors are characterized by thin structural panels.

Because the tower sizing is not included in this optimization problem, we limit the increase in the net rotor thrust at the given operating conditions to 14%, as done by Madsen et al. (2019) for the same rotor configuration. Given the lower loads generated
by the flexible baseline rotor, this constraint is more conservative than in the previous aerodynamic-only shape optimization study Madsen et al. (2019).

Finally, to ensure that there are no abrupt thickness changes between adjacent panels, we enforce linear adjacency constraints in the structural model so that two structural panels next to each other have less than a 5 mm difference in thickness for the current problems.

As mentioned in Sec. 4.1, the mass minimization optimizations enforce a torque constraint. Similar to the objective function, the constraint is normalized with respect to the baseline configuration performance.





## 4.4 Design Load Case

In this work, we limit our optimization studies to a single design point to demonstrate the framework's capabilities. We use the same reference below-rated-power design condition from the previous work on aerodynamic shape optimization (Madsen et al., 2019)—a uniform inflow velocity $V = 8\,\mathrm{m/s}$ and tip speed ratio (TSR)= 7.8. The same approach could be extended to a multipoint strategy (Kenway and Martins, 2015; Mangano and Martins, 2021; Madsen et al., 2019), averaging the turbine performance over 3 to 10 selected inflow cases.

In practical cases, fatigue and extreme load considerations must be used to size the rotor's internal structure. Although these loads are not considered in the present work, Caprace et al. (2022) have developed a multifidelity approach to address this shortcoming. The approach extends our high-fidelity approach to include life-cycle load considerations from conventional design tools.

## 4.5 Problem Summary

This section summarizes the optimization formulation and elaborates on the problem setup and numerical behavior. Table 1 lists the objective, variables, and constraints.

We allow the displacement of the outermost seven FFD sections shown in Figure 3, while the other sections are fixed. The twist derivatives on these fixed sections are flatter than their outboard counterparts. Enabling twist in this area harms the optimization convergence without noticeable performance benefits. The rest of the DVs are applied to the same outboard sections to ensure a fair comparison. The planform variables derivatives are scaled by a factor of 10 in pyOptSparse to speed up numerical convergence.

We set the SNOPT convergence tolerances for both the feasibility and optimality conditions to $10^{-6}$. While feasibility is quickly satisfied, optimality is slower to converge because of the high number of design variables and the design space nonlinearity. To limit the number of subsequent optimization restarts on HPC systems, we generally accept optimized layouts showing a plateau in the objective function and an optimality close to $10^{-4}$, yielding a decrease of more than two orders of magnitude with respect to the optimality in the baseline design. The computational cost and convergence are discussed in more detail in Sec. 5.

In this work, we do not perform dynamic analyses or include aeroelastic stability considerations in our formulation. However, flutter constraints have been implemented in the MACH framework (Jonsson et al., 2019b). As explained in Sec. 4.4, we consider a single-design-point optimization formulation. Nevertheless, the tools and the formulation above have already been extended to multipoint optimization studies (Madsen et al., 2019).





**Table 1.** Aerostructural optimization problem statement.

|  | Name | Symbol | Qty |
|---|---|---|---|
| Objectives | Torque | $Q$ | 1 |
|  | Mass | $M$ | 1 |
| Design Variables | Panel thickness | $\boldsymbol{x_{st}}$ | 117 |
|  | Twist | $\boldsymbol{x_{tw}}$ | 7 |
|  | Airfoil scale | $\boldsymbol{x_{sc}}$ | 7 |
|  | Chord | $\boldsymbol{x_{ch}}$ | 7 |
|  | Airfoil thickness | $\boldsymbol{x_{tk}}$ | 7 |
| Constraints | Stress | $\mathrm{KS}_\sigma \leqslant 1$ | 3 |
|  | Adjacency |  | 318 |
|  | Torque | $Q_x \geqslant Q_{x_{\mathrm{ref}}}$ | 1 |
|  | Thrust | $F_x / F_{x_{\mathrm{i}}} \leqslant 1.14$ | 1 |

Because of the assumptions on material properties for the structural model and the single below-rated inflow used in our studies, we do not start monolithic aerostructural optimizations from the benchmark turbine described by Bak et al. (2013). Instead, we use a loosely-coupled approach based on a sequence of structural optimizations to define a baseline structural layout. The resulting structural thickness distribution constitutes a more reasonable sizing for the inflow conditions reported above. This preliminary step, discussed in Sec. 5.1, reduces the overall computational cost. Moreover, it improves the robustness
of the subsequent coupled optimization, which has to traverse a smaller portion of the design space. We show how this latter approach is necessary to fully exploit the aerostructural trade-offs in the rotor design by comparing the loosely-coupled and monolithic, tightly-coupled optimizations.

## 5    Results

This section presents multiple optimizations. We first compare the loosely-coupled sizing approach to a tightly-coupled op-
timization strategy that optimizes only structural variables. We discuss the trade-off between computational cost and design performance for these two approaches. Then, we present several optimization studies to quantify the trade-off between rotor mass and torque for different design variables. Design trends over the span and blade sections are analyzed in detail, leveraging the blade-resolved model used by the CFD-CSM solver. Finally, we explore the effect of varying the optimization problem formulation on the rotor mass and torque.
As mentioned in Sec. 4.4, we optimize for a single inflow condition. Therefore, we focus our discussion on the tool capabilities and design trends rather than specific design parameters requiring broader life-cycle operating conditions. The three





different sets of design variables used for optimization—structural and twist variables (Tw), structural, twist, and scaled blade section variables (TwSc), and structural, twist, chord, and relative thickness variables (TwChTk)—are listed in Table 2.

**Table 2.** Design variable combinations for the optimization studies.

|  | Panel thickness | Twist | Airfoil scale | Chord | Airfoil thickness |
|---|---|---|---|---|---|
| Tw | ✓ | ✓ |  |  |  |
| TwSc | ✓ | ✓ | ✓ |  |  |
| TwChTk | ✓ | ✓ |  | ✓ | ✓ |

### 5.1 Mass minimization using uncoupled and coupled models

This first section explains the initial structural sizing obtained through a sequential, loosely-coupled aerostructural design strategy and compares it to the tightly-coupled MACH approach. The steps of this sizing process are illustrated in Algorithm 1.

---
**Algorithm 1** Loosely-coupled structural optimization

---
 1: Define initial layout
 2: **for** $i \leftarrow 0$ to $N$ **do**
 3:     1: Run MDA
 4:     2: Update aerodynamic input forces
 5:     3: Run structural optimization
 6:     4: Update structural layout
 7: **end for**

---

We run a sequence of structural optimizations and aerostructural multidisciplinary analysis (MDA) instead of a sole single-discipline optimization. In this way, we take into account the change in aerodynamic loads as the single-discipline optimization 405 alters the structural displacement. The first aerodynamic load distribution is obtained by running a coupled aerostructural analysis on a rotor with an arbitrary uniform thickness distribution. The optimizer minimizes the blade mass using this initial load while enforcing constraints on the aggregated stress. Once the structural optimization is completed, a new aerostructural MDA runs using the resulting optimized thickness distribution. The loads from the last MDA are passed to the structural solver for new structural optimization. We run a sequence of 13 MDA-optimization iterations until convergence. For this loosely-410 coupled approach, convergence means that any new structural optimization using an updated set of loads converges at the first (optimizer) iteration.

Because the optimization is purely structural, we cannot include geometric design variables and torque or thrust constraints in the optimization problem. The structural thickness distribution obtained through the loosely-coupled approach is suboptimal,





but it is a cheap and reasonable initial guess for the more expensive tightly-coupled problem. This preliminary step saves
hundreds of fully-coupled optimization iterations or time-consuming manual structural sizing. The mass and torque values
for the configuration we obtained serve the normalization values ($Q_{x_i}$ and $M_i$ in Sec. 4.1, and thrust $F_{x_i}$) for subsequent
tightly-coupled optimizations.

We now compare the results of the loosely-coupled sizing from Algorithm 1 to a subsequent MACH-based optimization.
Both optimizations minimize mass. The latter case uses structural thickness distribution from the loosely-coupled approach as
initial design. This problem includes the structural thickness variables and a torque constraint to ensure that the aerodynamic
performance does not degrade. The results of this study are presented in Table 3 and Figure 4

**Table 3.** The tightly-coupled optimization is more expensive but yields additional mass reduction.

|  | Uniform arbitrary | Loosely-coupled baseline | Tightly-coupled restart |
|---|---|---|---|
| Mass [kg] | 89,399 | 35,774 | 35,060 ($-2\%$) |
| Time [CPU-h] | – | 1,332 | 7,791 |

The initial structure with an arbitrary uniform thickness distribution is oversized for the inflow condition for which we
are sizing the turbine. The loosely-coupled optimization reduces the rotor mass from more than 89 tons to less than 36 tons.
The computation cost of this optimization is 1,332 core-hours on 160 Intel Xeon Platinum 8380 (Ice Lake) cores. This cor-
responds to less than 9 hours of computational wall time. Unless otherwise specified, the baseline case for all the subsequent
optimizations is the loosely-coupled optimization result.

The tightly-coupled optimization uses the full aerostructural model at each design iteration. The MDA and coupled deriva-
tives capture more subtle aerostructural trade-offs and further reduces the mass by 2%—roughly 700 kg for this layout. This
additional mass reduction is only possible through coupled optimization because the loosely-coupled approach could not reduce
the rotor mass any further.

Figure 4 shows where the optimizer reduces the mass over the blade. The view is perpendicular to the rotor plane. The
loosely-coupled optimum has the highest thickness over the spar caps away from the root and tip regions and on a large section
aft of the front spar on the lower skin (high-pressure side). The optimizer reduces this section's structural thickness in the
tightly-coupled optimization while reinforcing the upper skin counterpart. The area closer to the root shows thickness reduction
on both sides. Conversely, more material is added in front of the spar caps on both skins in the 50–80% span range. This added
material is due to the torque constraint. The optimizer has no direct control over the geometry in this design problem, so it
modifies the torque output by altering the blade elastic displacement and the bend-twist coupling through changes in the local
structural stiffness.

The specific layout modifications are load-case, parametrization, and constraint dependent. Our previous work showed that
the additional mass reductions achieved by the tightly-coupled model are consistent over different torque and displacement





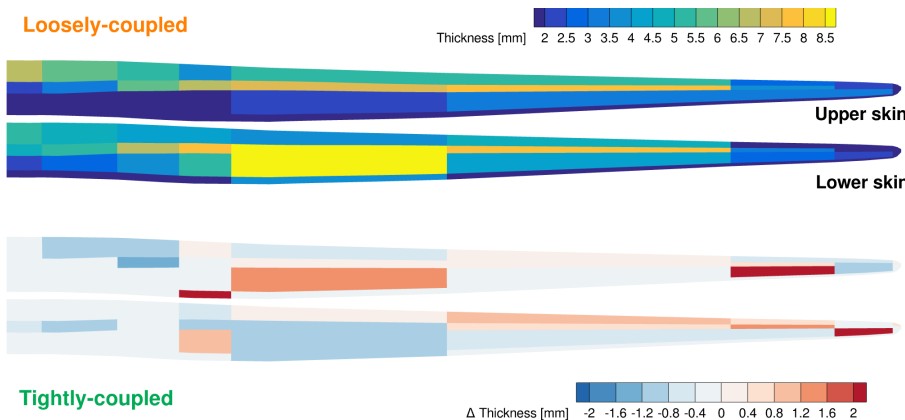

**Figure 4.** The optimizer adjusts the thickness to save 2% more mass in the tightly-coupled model compared to the loosely-coupled model.

constraint values and structural parametrizations. Recall that we are sizing the turbine for a below-rated inflow condition. In addition, we use a material with higher stiffness and comparable density properties compared to the fiberglass-based composites commonly used for industrial-scale wind turbines. Structural buckling is also not considered at this stage. These factors explain unrealistically low rotor structural thicknesses (1.6–8.5 mm). However, we expect the tightly-coupled optimization benefits

over the loosely-coupled approach to remain similar regardless of the material used.

### 5.2 Tightly-coupled aerostructural optimization

We now perform tightly-coupled aerostructural optimization including geometric design variables. The objective function is now the weighted average of torque and rotor mass defined by Equation (1). We use a weight of $\omega = 0.7$, emphasizing torque over mass. The effects of varying $\omega$ are discussed in Sec. 5.3.

Adding geometric variables yields higher performance improvements but increases the problem complexity and slows the optimization. As mentioned in Sec. 4, we set tight convergence tolerances for SNOPT. The optimizer consistently designs configurations that satisfy the linear and nonlinear constraints within thousands of one percent. However, as shown in Figure 5 for the case using structural and twist variables (Tw), it takes almost 300 design iterations to reduce the optimality by more than two orders of magnitude. Although the optimizer could further improve the design from a numerical standpoint, the mass

and torque are practically constant beyond 200 iterations. The variations beyond this point are likely smaller than the modeling error. The overall cost of this specific optimization is 25 000 core-hours. Future work on parametrization and tighter analysis convergence could address these robustness and efficiency issues, but the effect on the rotor design would be negligible from a practical design standpoint.

    We now discuss the optimized layouts. Figure 6 compares the baseline rotor and the Tw optimum. The rigid and operating

deflected shapes are included in both cases. The pitch sign convention is such that the zero-pitch position is at 90° relative to the incoming airflow, with the leading edge pointing upwards. The twist angle increases as the sections align with the freestream

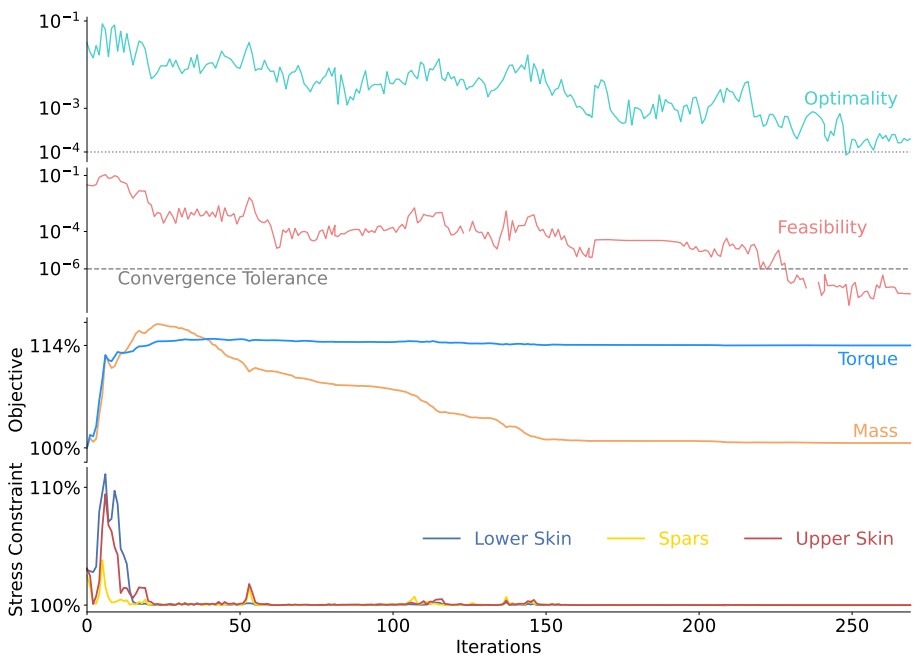

**Figure 5.** The Tw optimization converges to a feasible design in fewer than 300 iterations.

direction. The discontinuities in the plots originate from minor inconsistencies in the location of leading and trailing edge points at consecutive airfoil sections because we extract these distributions directly from the deflected aerodynamic meshes of the coupled solution.

The undeflected DTU 10 MW rotor outer mold line has a linear twist distribution (Madsen et al., 2019). The blade flexibility exacerbates the load-induced twist in the baseline design. On the one hand, the twist in the blade outboard half is higher than the rigid counterpart, which contributes to local load alleviation. On the other hand, the twist is reduced over the blade inboard half. These trends might be explained by the relative position of the elastic center and center of pressure at different spanwise locations, as the blade tapers from a circular section at the root to an FFA-W3 airfoil.

The optimizer decreases the twist by up to 6° between 45% and 85% of the span, increasing the aerodynamic loads in this section. Conversely, the tip unloads, and the local twist increases by up to 4°. The optimizer also adds a smaller 2° twist in the 20–40% span range, where the aerodynamic loads tend to re-align the local sections to the free stream. Because the optimized layout has minor structural changes (discussed in Figure 10), the twist deformation trend on the deflected blade is consistent with the one on the baseline rotor.

From an aerostructural perspective, a lower moment arm for the aerodynamic loads on the blade helps reduce the mass because of the lower root bending moment. However, it also reduces the in-plane moment that drives the torque generation. The optimizer identifies the best trade-off in this sense according to the function weights selected for the objective function.





The resulting design's torque is 14.3% higher at the given inflow condition. The structure thickens because of the higher loads, resulting in a mass increase of less than 1%.

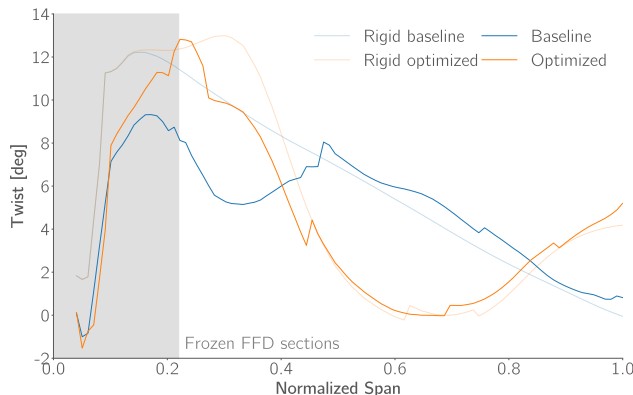

**Figure 6.** For the Tw optimum, the optimizer unloads the tip and reduces the twist between 45% and 85% of the span.

Next, we investigate how increasing design freedom benefits the optimized rotor performance. In Table 4, we compare the objective metrics for three optimizations with increasing levels of design freedom (see acronyms in Sec. 5). With the selected objective function and thrust constraint, the increase in torque is comparable over different optimizations. However, the optimizer utilizes the increased design freedom to reduce the mass by more than 9% for the TwChTk optimum.

**Table 4.** More design freedom yields similar torque improvements with increasing mass reduction compared to the baseline.

| DV set | Tw | TwSc | TwChTk |
|--------|--------|--------|--------|
| Torque | +14.3% | +14.3% | +14.6% |
| Mass   | +0.7%  | −6.3%  | −9.3%  |

     The twist, chord, and relative airfoil thickness distribution are compared in Figure 7. The twist distribution on the deflected

blades matches within 1–2° for the two optimizations that include planform design variables. The trend is comparable to the Tw case in the midspan section, but the twist increase is lower by up to 5° in the inboard section and up to 2 °the outer 20% of the blade.

     When the optimizer controls the blade planform, chord variations over the span compensate for these differences in twist. Both TwSc and TwChTk cases reduce the chord (and relative thickness for TwSc) by 30% close to the tip and increase it by

15% and 40%, respectively, between 22% and 45% of the span. The most complex case (TwChTk) couples the more aggressive chord increase with a decrease of up to 10% in relative thickness, stretching the airfoil in the chordwise direction. In contrast, the airfoil section gets shorter and thicker than the Tw and TwSc cases between 45% and 65% span, where the twist decrease is the highest.

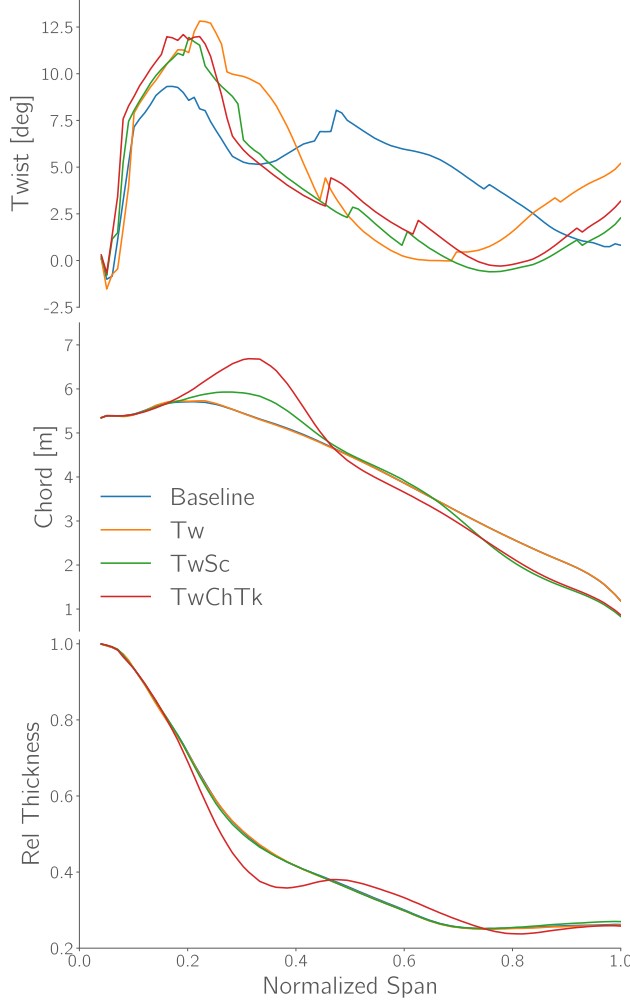

**Figure 7.** The additional design freedom from planform variables decreases the twist and chord at the tip compared to the Tw counterpart.

The effects of the structural and geometrical changes on the load distribution over the blade are shown in Figure 8. Both edgewise and flatwise force peaks shift inboard, reducing their moment arm. Forces in the span outermost 20% are lower than the baseline, but higher loads in the 20–80% span region compensate for the reduction of edgewise torque. The flatwise force peak is higher than the baseline but shifted inboard by 20% of the span. The corresponding integrated load does not exceed a 14% increase compared to the baseline because of the thrust constraint.

More design freedom enables more aggressive load shifts. The TwChTk case shows a marked load increase between 20% and 45% span as a consequence of the planform changes shown in Figure 7. The load shift is more significant for flatwise forces, which ultimately drive the blade sizing. This explains how the optimizer achieves the highest mass reduction in this



case. These trends could be specific to the selected inflow condition and rely on ADflow correctly capturing the aerodynamic loads at low speed near the root. However, these trends are consistent with engineering intuition.

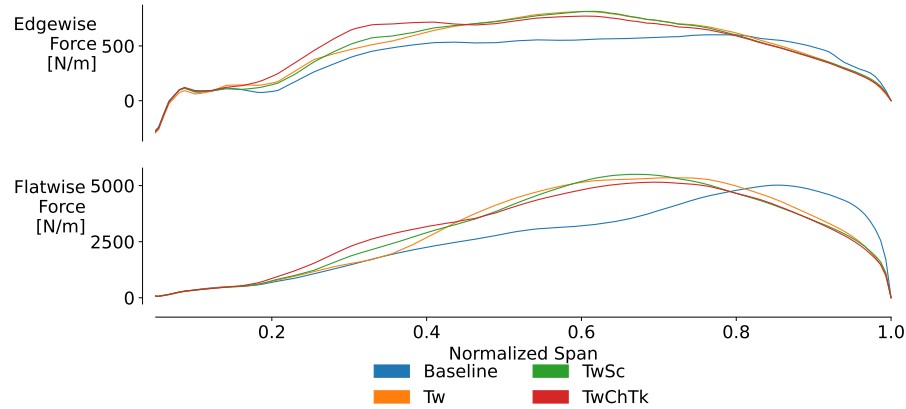

**Figure 8.** The optimization reshapes the spanwise load to move the peak inboard.

Thanks to MACH's high-fidelity analysis capabilities, we can investigate how the optimizer alters the local airfoil shape and pressure distribution, reported in Figure 9. The spanwise location of these sections is also highlighted in Figure 10. At 40% span, the $C_p$ over the normalized chord is comparable in peak location and magnitude between different optimized designs. The TwSc and TwChTk optimizations use planform variables to obtain this distribution with a lower twist than the Tw case, as shown in Figure 7. The TwChTk airfoil has a lower thickness-to-chord ratio and generates a higher pressure difference around the mid-chord.

At 60% of the span, the optimized layouts show a 50% increase in the suction peak compared to the baseline. They show similar $C_p$ trends despite the differences in local twist on the deflected blades. Unlike the previous case, the thickness-to-chord ratio for the TwChTk is higher, as highlighted in Figure 7. Closer to the tip, the $C_p$ distributions for TwSc and TwChTk are close to the baseline, and the force reduction shown in Figure 8 comes from the chord and thickness reduction. The Tw case has the same baseline planform and must reduce local lift through a more aggressive twist increase, which lowers the $C_p$ peak.

The two outermost baseline sections display a bump in the shape in the aft part of the airfoil, which results in a corresponding change in $C_p$. This bump appears due to the airfoil elastic deformation under the current loading condition. This is because of the thin baseline structural layout (obtained through the loosely-coupled approach), which reduces the chordwise stiffness. This issue can be addressed by enforcing stricter structural constraints.

Nevertheless, this phenomenon is interesting for two reasons. First, this level of detail is captured only through high-fidelity coupled analysis. Combined BEMT and beam codes cannot model this aerostructural interaction because the airfoil shape is fixed during analysis, and the airfoil drag polars do not account for chordwise deflections—only empirical corrections for 3D effects are available.

Second, the optimizer leverages this bump in the coupled aeroelastic solution. The optimizer removes the bump because it is detrimental to the aerodynamics (see 80% section in Figure 9). Conversely, the optimizer adds a small bump at the 60% section, 525 where the local load increases are the highest. However, the higher local torque contribution compensates for the effects of this local deformation. This detailed tailoring is only possible because our high-fidelity optimization approach accurately captures the two-way-coupled aerostructural elastic deformation.

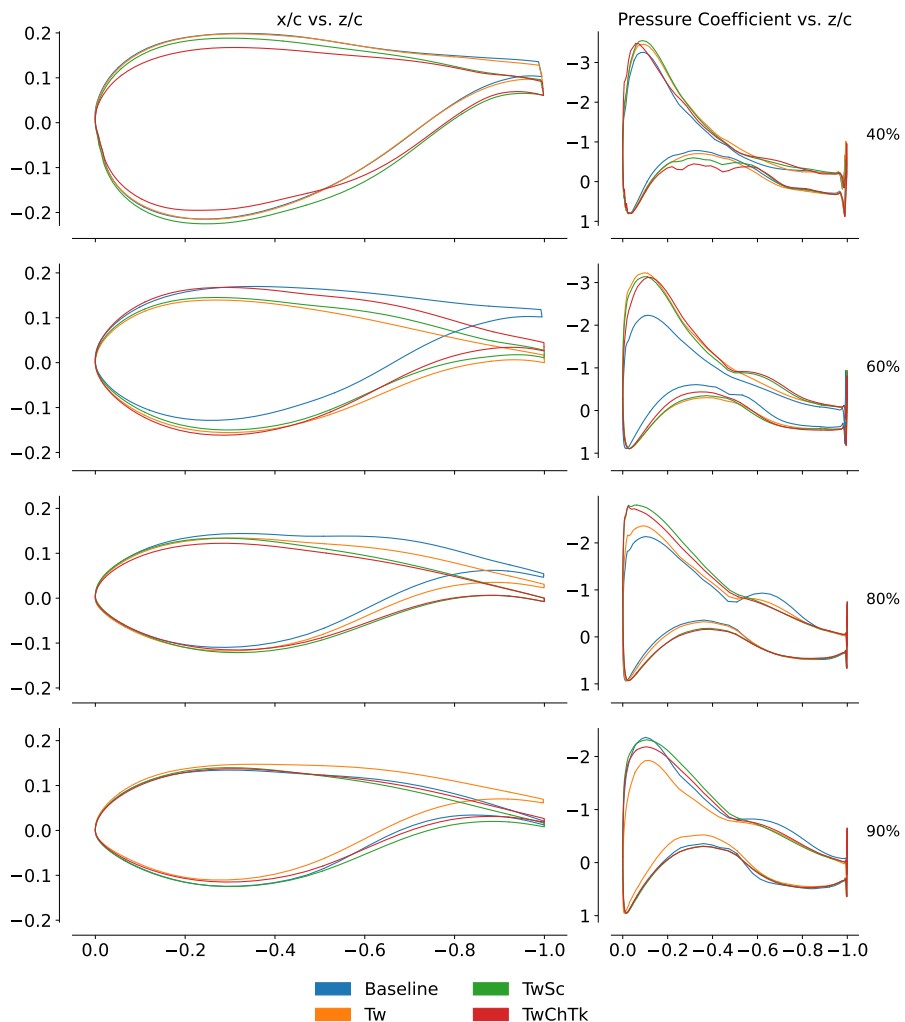

**Figure 9.** Cross-section pressure distributions show consistent optimization trends; when planform variables are optimized, the pressure peak at the leading edge increases. Section locations on the platform are highlighted in Figure 10.

To understand how the optimizer reduces the rotor weight in the different optimization cases, we compare the structural thickness distribution between the baseline layout and the three optimized configurations discussed in this section. In Figure 10,



we observe that the Tw optimization only made minor adjustments over the blade, for an increase of 242 kg over the whole rotor.

The planform variables enable a structural thinning over the thick lower skin panel, with reductions up to 3 mm for the TwChTk optimization. The thickness distribution trends are consistent between the TwSc and TwChTk over the rest of the blade. The main structural reinforcements occur at the leading edge section between the 80% and 90% of the span, most likely 535 to improve the bend-twist coupling and on the upper skin down to 30% of the span. The upper spar cap is also reinforced in both cases.

The changes on the cross-section made by the optimizer for the TwChTk case, in particular, affect both the local airfoil aerodynamics and the sectional bending stiffness. Because of the coupling of the aerodynamics and the structural response, it is hard to dissect which changes benefit the torque output and the mass reduction, assuming that a clear distinction exists. The 540 cases presented in this section exemplify the potential of our tightly-coupled high-fidelity approach to capture geometric and structural trade-offs. This approach can be extended to more elaborate design scenarios and rotor models.

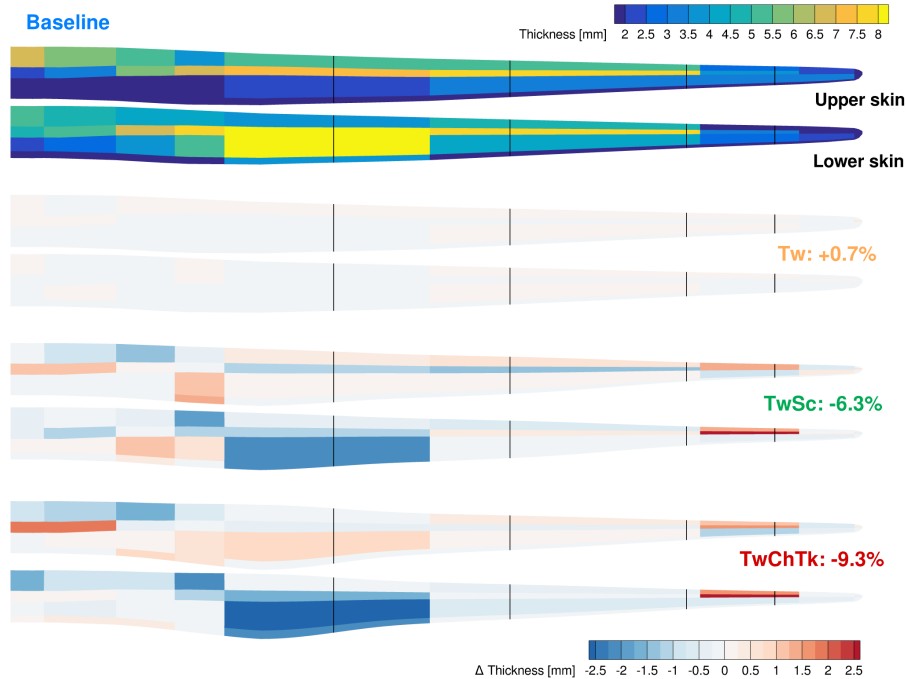

**Figure 10.** Increased geometric design freedom decreases the mass, particularly on the thick lower skin panel.

## 5.3 Design space exploration

Finally, we explore how the objective formulation affects the optimized design's torque output and total rotor mass. The results are summarized as a Pareto front over the mass-torque plane in Figure 11, where every point corresponds to a different





optimization case, color-coded to match the design variable sets discussed in Sec. 5.2. The designs closer to the lower right
       corner of the figure are lighter and more aerodynamically efficient, outperforming the other optimized layouts.

       The two cases in light blue (closest to the baseline) use a smaller subset of design variables and consider either only torque
       ($\omega = 1$) or mass ($\omega = 0$) in the objective function. The "Mass min" case is the tightly-coupled mass minimization problem
       discussed in Sec. 5.1. "Torque max" refers to a pure torque maximization problem ($\omega = 1$ in Equation (1)) using twist design
variables without changing the internal structural layout. The baseline rotor structure can tolerate only small load increases;
       thus, the optimizer increases the torque by no more than 4%.

       The case in the upper right corner of the figure (part of the orange Tw set) refers to another torque maximization case where
       the optimizer can change the twist and structural thickness distribution without mass penalty in the objective. The optimizer
       increases the torque by 18% without violating stress or thrust constraints. However, the total mass almost doubles because
the structural thickness increases to a point where the aerodynamic efficiency no longer improves with higher blade stiffness.
       Because this is not a relevant design case, we did not run TwSc and TwChTk cases with the same torque-only objective. These
       results reinforce the need to explore mass and torque trade-offs through objective function formulation.

       The remaining optimizations use both structural and geometric design variables. They also include the rotor mass in the
       objective. The cases ranging from 10 to 15% torque increase (see Figure 11) minimize Equation (1) using $\omega$=0.7, $\omega$=0.6, and
$\omega$=0.5 respectively.

       The rightmost cases in this set, using a 0.7-0.3 weighting for torque and mass, were discussed in Sec. 5.2. The mass reduction
       discussed in the previous section is consistent between different design variable sets, and the torque outputs are within 1–2%
       for the same objective weighting. Including planform variables reduces the mass by 6–8% compared to their Tw counterparts.
       Decoupling chord and thickness yields an additional 2–3% reduction. An objective with equally-weighted torque and mass
yields designs with a torque output between 9.4% and 10.5% higher than the baseline rotor and mass decreases of 5.4% to
       16.4%, depending on the parametrization.

       The relation between objective weights and final layout metrics becomes highly nonlinear when the objective emphasizes
       mass. With this formulation, the optimizer tends to reduce the aerodynamic loads more aggressively to achieve a lower mass.
       A case with $\omega = 0.4$ (not shown in Figure 11) had a torque output that is 10% lower than the baseline. For this reason,
optimizations exploring this design space area should prescribe the torque output rather than include it in the objective function.

       The two leftmost sets of optimizations in Figure 11 consist of mass minimization problems ($\omega = 0$ in Eq. 1) with a torque
       constraint that is either the same as the baseline or 5% higher. When the torque is increased by 5%, rotor mass increases
       by about 3% relative to the corresponding cases with baseline torque output. The load alleviation induced by the optimizer
       through the geometric design variables reduced the mass by 15% when only changing the twist distribution—a 13% additional
reduction relative to the design of Sec. 5.1.

       When the optimizer can scale the airfoil sections, the mass decreases by an additional 7%. In the scenario with the highest
       design freedom with decoupled chord and relative thickness variables, the optimizer achieves a 27% lighter structure than
       the baseline while maintaining the same torque output at the given load condition. Our optimization framework can explore





more objective function combinations and rotor parametrizations. Such exploration would be impossible with gradient-free approaches because the computational cost would be intractable.

## 6  Conclusions

We presented the first aerostructural optimization study of a wind turbine rotor using a coupled CFD-CSM solver and an efficient gradient-based optimization strategy. The developed framework can optimize rotor mass, torque output, or a combination of the two by modifying the structural thickness distribution and the blade outer mold line shape. A preliminary sizing approach based on a conventional loosely-coupled approach leverages the blade-resolved 3D structural model without using the more accurate but expensive tightly-coupled model. The tightly-coupled model reduces the mass by 2% more than the loosely-coupled approach for a fixed blade geometry. This improvement can only be achieved through the aerostructural solver and its coupled derivatives.

Changes in the twist, chord, and relative thickness distribution reduce the mass by up to 9% and increase the torque by up to 14%. The flatwise and edgewise loads are moved away from the tip to find the best compromise between aerodynamic efficiency and structural sizing. The airfoil layout and pressure distribution over the blade are also discussed in detail, contributing insights that cannot be obtained with conventional design methods.

Finally, the Pareto front analysis highlights optimal rotor mass and torque output trade-offs. The benefits of more complex geometrical parametrization are consistent over the design space. The decoupling of chord and relative thickness reduces the mass by 27% without reducing the initial torque output.

The results showcased in this paper open the door to the industrial application of high-fidelity optimization to the design of the next generation of wind turbines. To achieve more practical results, we could use a more detailed structural model and implement more wind conditions for performance and loads. An ongoing effort is coupling conventional codes with MACH to inform fatigue and extreme-loads constraints with loosely-coupled dynamic simulations.

*Author contributions.* Marco Mangano searched and analyzed literature sources, setup and verified the aerostructural model, planned the case studies, setup and ran the analyses and optimizations, postprocessed and interpreted the results, and wrote most of the paper. Sicheng He contributed to the structural mesh generation and analysis setup, the methodology section details, and was actively involved in debugging the aerostructural solver and its coupled derivatives. Yingqian Liao supported the solver and optimization setup and tuning, the study planning, and the interpretation of the results, providing expert feedback on the fluid-structure interaction. Denis-Gabriel Caprace provided thorough feedback on the case study planning and results discussion on wind turbine design. Andrew Ning provided expert feedback on wind turbine design optimization for the results discussion. Joaquim R. R. A. Martins has been the project advisor, secured funding and computational resources, and advised on the optimization setup, results discussion and visualizations. All authors reviewed and edited the paper.



**Figure 11.** Mass and torque trends for different design variable sets are consistent over the design space.



*Competing interests.* The MACH framework, co-invented by Joaquim Martins, is used by Supercritical Research, LLC, a company owned by Joaquim Martins.

*Acknowledgements.* The authors would like to thank Mads Madsen and Frederik Zahle for sharing the FFD grids, aerodynamic meshes, and baseline geometry input for PGL. Madsen's tips were helpful in ensuring pyGeo was correctly handling the blade shape modifications. Thanks also to Anil Yildirim for helping with ADflow tuning to increase the solver's robustness and speed. This research was supported by the Department of Energy (DOE) Advanced Research Projects Agency-Energy (ARPA-E) Program award DE-AR0001186 entitled "Computationally Efficient Control Co-Design Optimization Framework with Mixed-Fidelity Fluid and Structure Analysis". The authors thank DOE
ARPA-E Aerodynamic Turbines Lighter and Afloat with Nautical Technologies and Integrated Servo-control (ATLANTIS) Program led by Mario Garcia-Sanz. Special thanks to the entire ATLANTIS Team for their support. This research was partly supported through computational resources and services provided by Advanced Research Computing at the University of Michigan, Ann Arbor. The authors also used the Texas Advanced Computing Center (TACC) Stampede2 High Performance Computing system via the Extreme Science and Engineering Discovery Environment (XSEDE), which is supported by the National Science Foundation grant number ACI-1548562.





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
