# Peer review of "Aeroelastic Tailoring of Wind Turbine Rotors Using High-Fidelity Multidisciplinary Design Optimization"

_Wind Energy Science, 2023_

## Referee Comment (RC1)

**Review of WES-2023-10**

The study presents a tightly coupled, 3D optimization framework based on FEM analysis and CFD. The method has a lot of potential and if properly fine-tuned can be used to greatly improve the current state o the art when it come to wind turbine rotor design.

The work is very interesting, and this line of work has a lot of potential. Nevertheless, some of the details of this work raise some questions in my opinion. Firstly, the scope of this work is not clear: is it intended as an illustration and first showcasing of the coupled optimization method? If so important details regarding the method's details are missing throughout the paper, and it would be very hard for a third party to reproduce the results showcased with the information contained in the paper. On the other hand, if the objective is to discuss the results of the optimization discussed in the paper, various corners appear to have been cut: the blade is made of aluminum and no details regarding how the distribution along the blade of the various thickness panels is chosen is given. From an aerodynamic perspective, no mesh convergence & important details on model set-up are presented, and baseline results are not compared to other author's predictions for this testcase. Moreover, the single-point optimization, without accounting for extreme loads in other operating and parked conditions is questionable, and the authors also acknowledge this in the paper.

Title: "Aeroelastic Tailoring": What is the reasoning for including this term in the title? Perhaps consider elaborating on the concept of aeroelastic tailoring and what it means in the context of this study when discussing results in section 5.2.

Literature review: An interesting concept that I don't think was considered in the literature review is to use high-fidelity simulation to train a meta-model such as an Artificial Neural Networks (ANN) to perform design exploration, such as proposed by (Lorenzo Cozzi *et al* 2022 *J. Phys.: Conf. Ser.* **2265** 042050). The best candidate designs can then be simulated, and the ANN can be updated and the process repeated if needed.

Figure 1: This illustration is very detailed, but it is hard to read. My suggestion is to move it to an appendix and focus on a more streamlined and simple illustration here. Moreover, the differences between the loosely-couple and tightly-couple aero structural optimization loops should be investigated.

Section 3: No details regarding boundary conditions, problem formulation, turbulence model and numerical domain are given. Moreover, authors state that various meshes are tested, but no comparison between them is presented. The choice of L1 mesh makes sense to reduce core-hours but does it ensure high enough accuracy?

Section 3: It is not clear to me how the structural problem is formulated. Is it a static analysis? Is it possible to account for complex aeroelastic interactions with the methods (flutter, vortex-induced vibration, etc…)

L463: ". The discontinuities in the plots originate from minor inconsistencies in the location of leading and trailing edge points at consecutive airfoil sections because we extract these distributions directly from the deflected aerodynamic meshes of the coupled solution" – I think this needs to be explained better. Without further information I would not expect differences in local deformation to lead to discontinuities in twist as seen in Figures 5 and 6.

Figure 6: It is not clear to me what is being shown here. What is the difference between the "Rigid" and flexible cases? Is it the difference caused by the blade deflection or is it a different starting geometry?

---

## Author Comment (AC1)

**Aeroelastic Tailoring of Wind Turbine Rotors Using High-Fidelity Multidisciplinary Design Optimization**
**Wind Energy Science**
**Manuscript ID: WES-2023-10**

Marco Mangano, Sicheng He, Yingqian Liao, Denis-Gabriel Caprace, Andrew Ning, and Joaquim R. R. A. Martins

**Community Comments and Response**

**Comment 1:** [...] Maybe you could extend the sentence about Horcas et al. (2022) slightly to also reflect that different BEM model implementations can generate different results, and that models covering a range of aerodynamic fidelities have been evaluated in Horcas et al (2022)?

**Response:** Thank you for the insightful comment, this is an important clarification on modeling consistency. We expanded our comment on Horcas et al. (2022) to mention the broader scope of the paper and the impact of implementation differences in BEM results.

The manuscript changes to address this comment are highlighted in green.

---

## Author Comment (AC2)

**Aeroelastic Tailoring of Wind Turbine Rotors Using High-Fidelity Multidisciplinary Design Optimization**
**Wind Energy Science**
**Manuscript ID: WES-2023-10**

Marco Mangano, Sicheng He, Yingqian Liao, Denis-Gabriel Caprace, Andrew Ning, and Joaquim R. R. A. Martins

**Reviewer 1 Comments and Response**

Thank you for your feedback on our work. We break down and address your comments below. Note: Actions taken to address the reviewers' comments are highlighted in red.

The corresponding changes in the manuscript are highlighted in red in the manuscript. Some of the responses to reviewer 2, highlighted in yellow, are also relevant to these comments. Responses to the community comments are highlighted in green.

**Comment 1:** Firstly, the scope of this work is not clear: is it intended as an illustration and first showcasing of the coupled optimization method? If so important details regarding the method's details are missing throughout the paper, and it would be very hard for a third party to reproduce the results showcased with the information contained in the paper.

**Response:** The reviewer correctly identifies the dual scope of this work. We agree that this should be made more explicit in the text. Necessary clarifications have been added to Sect. 1.2.

Regarding the replicability of the results, both the wind turbine model and the framework we use have been extensively described in detail in the literature. Although we did not upload the input files of our model on a public repository, we would be happy to share them with the interested reader. A sentence about data availablility has been added at the end of the manuscript. The benchmark turbine we use as a reference is available on the official DTU GitLab repository. We discuss the assumptions of our model and the differences from the original configuration in Sect. 3. The publicly available components of the MACH framework are referenced in the footnotes, and several literature references are provided for individual components and overall framework applications. Moreover, NASA's fully open-source MPhys framework provides comparable aerostructural optimization capabilities using both MACH solvers and other well-known CFD tools. Considering that the definition of the benchmark we studied in this work is publicly available, and the open character of the software we used, we believe this study is replicable.

**Comment 2:** On the other hand, if the objective is to discuss the results of the optimization discussed in the paper, various corners appear to have been cut: the blade is made of aluminum and no details regarding how the distribution along the blade of the various thickness panels is chosen is given

**Response:**

For the distribution of the various thickness panels, we subdivided the blades following the main geometrical

features of the planform. We further clarified the process in Sect. 3.

We acknowledge that our model has modeling limitations and we are actively developing an updated structural model that will address these concerns. We aim at presenting new results for aerostructural optimization using anisotropic composite materials at the WCSMO 2023 conference and a following journal publication.

The assumption of isotropic material properties is necessary at this stage of development. Aluminum properties have been selected for convenience and can be changed at runtime in our scripts. We stress again that this is a demonstration of the capabilities of the tool at the current stage, and several previous works with MACH have discussed optimizations of composite wingboxes [1].

The ongoing effort mentioned above will also use a more refined parametrization both along the span and the blade section.

**Comment 3:** From an aerodynamic perspective, no mesh convergence & important details on model set-up are presented, and baseline results are not compared to other author's predictions for this testcase.

**Response:** The aerodynamic mesh and solver are the same as the ones used by Madsen et al. [4]. That work presents a grid convergence study and a comparison with DTU's Ellypsys CFD solver, extensively discussing the pros and cons of the two tools. We ensured that our aerodynamic simulations match the values from the previous study, but chose not to add a section for the sake of conciseness. As more readers might be looking for the same information, we extended the discussion to redirect them to Madsen et al. [4] for more details.

**Comment 4:** Moreover, the single-point optimization, without accounting for extreme loads in other operating and parked conditions is questionable, and the authors also acknowledge this in the paper.

**Response:** We acknowledge that the sizing and design of this turbine are not to be intended as a reference for practical applications, as we focus on the capability demonstration of our code. We explicitly investigate the performance of the blade at below-rated conditions as our framework does not include turbine controls or dynamic simulation capabilities.

Academic work and industry practice demonstrated how single-point optimizations exploit geometrical and structural features for the specific design point, often at the detriment of the performance over the rest of the operational envelope [3]. Two separate works will address this limitation. On the one hand, we are investigating a different multipoint optimization problem formulation that includes site-specific considerations. We aim to present the study at the WESC 2023 conference and a following publication. On the other hand, we are actively working to extend the study started in [2] to enable constraints for extreme and fatigue loads in our high-fidelity optimization framework. Results will be discussed in a separate publication.

We made these points more explicit in Sects. 1.2 and 4.4.

**Comment 5:** Title: "Aeroelastic Tailoring": What is the reasoning for including this term in the title? Perhaps consider elaborating on the concept of aeroelastic tailoring and what it means in the context of this study when discussing results in section 5.2.

**Response:** We agree on this point. Aeroelastic tailoring is now discussed in Sect. 5.2 and 5.3

**Comment 6:** Literature review: An interesting concept that I don't think was considered in the literature review is to use high-fidelity simulation to train a meta-model such as an Artificial Neural Networks (ANN) to perform design exploration, such as proposed by (Lorenzo Cozzi et al 2022 J. Phys.: Conf. Ser. 2265

042050). The best candidate designs can then be simulated, and the ANN can be updated and the process repeated if needed.

**Response:** The authors are aware of the mentioned paper, which presents a promising methodology in the field of wind turbine design optimization. However, we feel that such a methodology differs significantly from what we propose in our work. A more extensive literature review including optimization through meta-models would increase the length of this manuscript without adding information useful for the reader to better frame our research. Therefore, we mention the work in Sect. 1.1 but we choose to leave further discussion out of the literature review presented in this manuscript.

**Comment 7:** Figure 1: This illustration is very detailed, but it is hard to read. My suggestion is to move it to an appendix and focus on a more streamlined and simple illustration here. Moreover, the differences between the loosely-couple and tightly-couple aero structural optimization loops should be investigated.

**Response:** We agree on the need to make the XDSM diagram easier to read. We removed unnecessary components from the diagram and increased the size of the figure. We also refer directly to the XDSM components in the framework description in Sect. 2 Moreover, we added clarifications in Sect. 5.1 to help the reader understand the differences between the two approaches. As for the loosely coupled approach, we added a simplified XDSM diagram to the manuscript. Together with Algorithm 1, it should help to clarify the differences between the two optimization strategies.

**Comment 8:** Section 3: No details regarding boundary conditions, problem formulation, turbulence model and numerical domain are given. Moreover, authors state that various meshes are tested, but no comparison between them is presented. The choice of L1 mesh makes sense to reduce core-hours but does it ensure high enough accuracy?

**Response:** We use the Spalart–Allmaras turbulence model for this work, as mentioned in Sect. 2. The details of the boundary conditions, problem formulation, and numerical domain are detailed by Madsen et al. [4] We added some high-level details and explicitly directed readers to that previous work for this information. This previous work also details a thorough grid convergence study and uses both L1 and L0 meshes. As discussed by Madsen et al. [4], ADflow overpredicts the loads on the L1 meshes; but preserves the same features and load trends. The comparison of baseline and optimized layouts in Fig. 1 highlights this behavior. This figure was not added to the manuscript for sake of conciseness. Therefore, we expect the results of an optimization with L0 to be consistent with those based on L1. With the resources available for this study, an optimization with L0 was numerically untractable.

[Figure]

Figure 1: Aerodynamic loads for the baseline and the optimized rotor. For each rotor, the load distributions are computed with the L1 and the L0 mesh. Both the normal and the driving forces are overpredicted with the L1 mesh, but the load distributions remain consistent with the L0 results.

**Comment 9:** Section 3: It is not clear to me how the structural problem is formulated. Is it a static analysis? Is it possible to account for complex aeroelastic interactions with the methods (flutter, vortex-induced vibration, etc...)

**Response:** We run a static analysis of the structural model under steady aerodynamic loads, so dynamic instabilities are outside the scope of this study. We clarified the structural setup in Sect. 3

**Comment 10:** L463: ". The discontinuities in the plots originate from minor inconsistencies in the location of leading and trailing edge points at consecutive airfoil sections because we extract these distributions directly from the deflected aerodynamic meshes of the coupled solution" - I think this needs to be explained better. Without further information I would not expect differences in local deformation to lead to discontinuities in twist as seen in Figures 5 and 6.

**Response:** We updated the explanation in the manuscript. The noise in the twist distribution is not due to local deformation, but to a non-deterministic behavior when ADflow identifies the surface mesh nodes associated with the leading and trailing edges. From one section to the next, the locations identified as LE and TE can switch between the discrete collection of airfoil mesh nodes. Since the twist is defined as the angle between a reference line and the chord line (connecting LE and TE), even small perturbations of these points lead to the discontinuities shown in the plot. This limitation in ADflow postprocessing will be addressed in future work.

**Comment 11:** Figure 6: It is not clear to me what is being shown here. What is the difference between the "Rigid" and flexible cases? Is it the difference caused by the blade deflection or is it a different starting geometry?

**Response:** We updated the manuscript and the caption to clarify this point. We now use "deflected" and "undeflected" to avoid confusion. "Rigid" refered to the undeflected blade shape on which the geometry deformations are applied. The other case refers to the same blade deflected under the aerodynamic loads at the prescribed inflow conditions we use in our optimization. This figure displays how the twist distribution

changes when loads are applied to the blade, highlighting how the optimizer accounts for passive load alleviation in the design process.

**References**

[1] Timothy R. Brooks, Joaquim R. R. A. Martins, and Graeme J. Kennedy. Aerostructural trade-offs for tow-steered composite wings. *Journal of Aircraft*, 57(5):787–799, September 2020.

[2] Denis-Gabriel Caprace, Adam Cardoza, Andrew Ning, Marco Mangano, Sicheng He, and Joaquim R. R. A. Martins. Incorporating high-fidelity aerostructural analyses in wind turbine rotor optimization. In *AIAA SciTech Forum*, January 2022.

[3] Mark Drela. *Frontiers of Computational Fluid Dynamics*, chapter Pros and Cons of Airfoil Optimization, pages 363–381. World Scientific, Singapore, November 1998.

[4] Mads H. Aa. Madsen, Frederik Zahle, Niels N. Sørensen, and Joaquim R. R. A. Martins. Multipoint high-fidelity CFD-based aerodynamic shape optimization of a 10 MW wind turbine. *Wind Energy Sciences*, 4:163–192, April 2019.

---

## Author Comment (AC3)

**Aeroelastic Tailoring of Wind Turbine Rotors Using High-Fidelity Multidisciplinary Design Optimization**
**Wind Energy Science**
**Manuscript ID: WES-2023-10**

Marco Mangano, Sicheng He, Yingqian Liao, Denis-Gabriel Caprace, Andrew Ning,
and Joaquim R. R. A. Martins

**Reviewer 2 Comments and Response**

Thank you for your feedback on our work. We break down and address your comments below. Note: Actions taken to address the reviewers' comments are highlighted in red.

The corresponding changes in the manuscript are highlighted in yellow in the manuscript. Some of the responses to reviewer 1, highlighted in red, are also relevant to these comments. Responses to the community comments are highlighted in green.

**Comment 1:** The numerical framework is described in several other publications and this new article seems more focused on results. However I struggle to extract takeaways that can be applied to the real world.

**Response:** We agree that more emphasis could be put on the takeaways from the paper. A similar point was made by Reviewer 1. We clarified the scope in Sect. 1.2. This is the first time the framework is referred to in the wind energy community. This is why we deemed it appropriate to re-establish the workflow and present the main components. The reader is referred to other publications for the specific details of MACH framework, which we omitted in this paper for the sake of conciseness.

**Comment 2:** First, blades are modeled as made of Aluminum, which is clearly not the case for real blades. This to me means that little to nothing of the structural results can be applied to real blades made of composites.

**Response:** As we clarified in our answer to the previous comment, the intent of the paper is not to provide results that can be directly applied to real blades. Our main focus is on the tool's capabilities. We made these limitations and the overall scope of the paper clearer in Sect. 1.2, Sect. 3, and Sect. 5

There is nothing in the framework that limits its utilization to isotropic blades, as demonstrated in previous works based on MACH. The use of aluminum for the blade was chosen as a necessary simplification to restrict the scope of an already long paper.

An ongoing research effort is focusing on developing a more refined structural model and extending the optimization studies to leverage the anisotropic properties. These studies will be the object of upcoming conference presentations and publications.

**Comment 3:** If we stick to aerodynamics, I fear that the proposed design solutions violate standard stall margins, especially along span.

**Response:** We understand the concern. Previous works with MACH included a separation constraint to prevent stall at high angles of attack. However, the aerodynamic solutions obtained by ADflow do not suggest that any separation is occurring over the blade—see the $C_p$ distributions in Fig. 10 (revised manuscript). Additionally, we added Fig.12 to discuss streamlines, pressure and stress distribution over the 3D model of one of the optimized layouts.

We deem a tighter bound on twist design variables (as a proxy to impose a stall margin constraint) here would also be an unnecessary limitation for the current study. Nevertheless, we have analyzed the optimized pitch distribution and mention the largest changes in the discussion for Fig. 8. Using an unmodified twist distribution compared to the baseline, Madsen et al. [3][Sect. 6.1] found an approximately 6.5 degree pitch as optimal to increase torque at the same given inflow condition. Since the blade did not stall at this new condition, this gives an idea about how far the original design is from stall. We do not consider pitch changes at this below-rated condition, but the local twist changes in our optimized designs do not exceed the values from the previous work.

We are aware that in a real application (which is again beyond the scope of these academic optimization studies) stall might occur in this configuration. This comment rather highlights another possible limitation of the current tool in realistically predicting stall.

On the one hand, the flow is assumed fully turbulent, which would delay flow separation on a real blade. On the other hand, RANS codes are known to over-predict $C_{L_{max}}$ and the stall angle of attack. We added a paragraph in Sect. 5.2 to discuss these points.

Regardless of the solver specifics, the optimizer would avoid a blade design affected by stall as it would greatly hinder the blade performance. Single-point optimization exacerbates design features (such as, for example, pointy leading edges or aggressive twist distributions) that are effective in a narrow range of operating conditions while possibly being detrimental at other flow conditions. This is the main motivation beyond an ongoing research effort on multipoint optimization that will go beyond the cases discussed by Madsen et al. [3] for aerodynamic shape optimization.

**Comment 4:** Overall, I really miss a comparison against lower fidelity conventional tools. Was the whole high-fidelity framework really necessary to identify these tradeoffs? Why wouldn't the same tradeoffs emerge at lower fidelity?

**Response:** We agree that a comparison with lower fidelity tools —even for analysis only— would be crucial to understand how to exploit these different tools during the design process. MACH does not have the capabilities or scope to replace these tools, but should rather be used to integrate the shortcomings of the former during a preliminary design stage. The mixed-fidelity approach we are further developing combines MACH and WEIS, so we could leverage the latter to understand the pros and cons of the purely high-fidelity approach [2]. We aim to provide a detailed comparison of the two tools on the reference and optimized designs. At the current stage and to the best of our knowledge there is no equivalent study performed with lower fidelity tools with a comparable problem formulation. A one-to-one comparison with a tool like WEIS would be beyond the scope of the current investigation.

The use of a blade-resolved 3D model in the optimization has two main advantages. On the one hand, despite the limitations of the aerodynamic model discussed for comment 3 above, the coupled analysis can capture spanwise aerostructural interactions without assumptions or empirical corrections on the flow behavior and local stiffness properties. The Fig. 12 we added highlights the additional insights available to the designer. The low fidelity method would likely fail to capture some of the changes in shape that we identified when the blade gets loaded (e.g., the "bump" discussed in Fig. 10), and the feedback they have on aerodynamics.

On the other hand, using 3D geometrical deformations and panel thicknesses as design variables enables the optimizer to explore a broader and more realistic design space while again relaxing assumptions on condensed mechanical properties of a beam, and limitations associated with the use of a discrete collection of airfoils with tabulated aerodynamic properties. This again advocates for a strongly-coupled monolithic approach which is best tackled using high-fidelity methods.

We discuss the points above in the newly added Sect. 4.6

**Comment 5:** As a reader, I'm left with the strong feeling that the large mass reductions are too good to be true and only a numerical artifact generated by the unresolved weaknesses of this novel framework. Can the authors prove me wrong?

**Response:** We understand the concern about the large mass reduction. We summarized the points below in Sect. 5 and Sect. 6 of the manuscript. The mass savings seem unrealistic or just not practical because of a couple of assumptions in the studies. The first one is the overly conservative baseline model sized through the loosely coupled approach for the specific below-rated condition. We discuss the limitations of this approach in Sect. 5.1. The second is the absence of extreme load conditions sizing and fatigue considerations. We have clarified these major assumptions in the revised manuscript.

With the above assumptions, a large mass reduction is possible and reasonable. First, we observed an inboard load shift in the optimized layouts, which led to lower bending moments and hence allowed for thinner blade structures. Enabling shape change helps redistribute the load and ultimately contributes significantly to mass reductions. Additionally, we have further performance gains because the baseline model that we adopted is, from a chord and twist distribution standpoint, a suboptimal variant of the original benchmark model [3]. The model's suboptimal performance provides even more room for performance improvements.

As for the concern of numerical artifacts being exploited by the optimizer, the structural solver has been extensively verified in previous work and we are confident that the structural stress constraint is appropriately accurate for this design study. The scenario where the absence of a buckling analysis affects the mass gains does not apply to the studies we are presenting. The largest mass gains (shown in Fig.11) come from the thinning of a large panel on the lower skin, which is loaded in tension.

Nevertheless, the objective of this work is to highlight how our high-fidelity MDO tool can explore design space and present preliminary studies. We are further developing our tools and models. We will present more practical studies in upcoming journal submission and conference presentations.

**Comment 6:** Page 1 Abstract: It would be nice to learn how the optimizer achieves such large gains already in the abstract

**Response:** Thanks for the suggestion. We added more details in the abstract.

**Comment 7:** Page 3 Line 53-60: The whole first paragraph can be safely deleted. No need to talk about climate change in a wind energy journal

**Response:** We see how we are "preaching to the choir" with this introductory paragraph. However, we hope the paper targets a broader audience and we feel that the sources we provide are useful regardless of the readers' expertise on the subject. For this reason, we would prefer to keep this introductory paragraph but we rephrased it to reduce redundancy.

**Comment 8:** Page 3 Line 65: lower solidity does not necessarily mean higher efficiency. For modern WTs,

lower solidity mostly means higher TSR and lower drivetrain costs

**Response:** Thanks for the insightful comment, we edited the sentence accordingly.

**Comment 9:** Page 4 Line 108: OpenFAST is not really based on a multibody dynamics model

**Response:** Thanks for highlighting this inconsistency, we updated the description

**Comment 10:** Page 4 Line 113: A reference seems missing. Maybe some publications from IEA Wind TCP Tasks 29 or 47 could be used

**Response:** Thanks for pointing us to relevant sources, we add citations to this claim.

**Comment 11:** Page 5 Line 120: I would add "unless coupled to an airfoil solver"

**Response:** We edited the sentence accordingly.

**Comment 12:** Page 10 Line 252: the fact that blades are made of Aluminum is a major limitation of this study. This shortcoming cannot be buried in page 10 or many readers will miss it.

**Response:** We agree on the fact that the model limitations need to be highlighted earlier in the manuscript. We made changes to Sect. 1.2 to make it more explicit.

**Comment 13:** Page 13 Line 344: Why 14% more thrust? +14% in max thrust would be a big increase. At the same time +14% in region II might not be an issue as long as the peak thrust shaving logic is applied when thrust reaches its maximum. Overall more discussion around this assumption seems needed.

**Response:** We agree on the need for clarifications and we now discuss more details in Sect. 4 The value we use is consistent with the one used by Madsen et al. [3]. We refer specifically to the design point we use, which is at below-rated conditions (region II). We do not model peak shaving, but assume the controller can handle loads at rated power conditions. An extension of Caprace et al. [2] could compare power profiles over regions II and III using WEIS on a consistent low-fidelity model.

**Comment 14:** Page 14 Line 355: In the report DTU Wind Energy Report-I-0092 rated TSR is 7.5, not 7.8. Who is wrong?

**Response:** We refer to and use the same verified models of [3], so we kept the same tip speed ratio used in that work. This is now clarified in the manuscript. Moreover, the modifications made by Madsen to the baseline rotor by Bak [1] would justify the change in the TSR in region II. The ongoing work focused on multipoint optimization will explore a broader range of wind and TSR conditions.

**Comment 15:** Page 20 Figure 6: Nowhere in the paper I see the word stall, which is however a key consideration in real wind turbine blade design. Designers must enforce a minimum margin to stall, or the turbulent inflow will keep the blade in and out of stall. If you had this constraint, I doubt you'd be able to drop twist so much in the mid span. Please check the operating angles of attack of the airfoils and how close you are to the stall point

**Response:** We understand the concern, and we try to address it in the response to comment 3 above. We believe that, among other limitations of the solver, this is a characteristic phenomenon of single-point optimizations. We now explicitly mention the absence of stall constraints in Sects. 4 and 5, so that experts can better understand how these large changes in the angle of attack are possible.

ADflow has the capability to handle separation constraints. However, such constraints would not be active for the designs we are exploring due to the considerations we made above. We could include a dedicated design point in the multipoint stencil to specifically address the stall margin.

For what concerns the local angle of attack, we now explicitly mention the maximum pitch increases in Sect. 5.2. Our reply to comment 3 should address other related concerns.

**Comment 16:** Page 21 Figure 7: Max chord is often constrained. Here different solutions have different max chord values. Is this a fair comparison? Why doesn't the optimizer favor larger chords?

**Response:** Thanks for highlighting the missing information on geometrical variable bounds, we now discuss them in Sect. 4 and Sect. 5.3.

We enforce an upper bound of 40% increase of planform design variables (chord, thickness, and the "scaled airfoil") with respect to the initial geometry. This bound is only active at a single section for the TwChTk case, where the optimizer can modify the chord and relative thickness independently. In this case, the chord is effectively limited to approximately 7 m at the largest location, which is reasonable compared to industry standards for large turbines. Again, we stress the fact that the optimized layouts should not be considered as practical designs due to the model and optimization assumptions at the current development stage.

As for your concern on the fairness of the comparison, the three optimized solutions refer to different design variable setups and we are presenting this comparison to specifically highlight and quantify the impact of adding a "scaled airfoil" (TwSc) and decoupled chord and relative thickness (TwChTk) variables on top of the simplest, twist-only case. The differences in the optimized planform arise from the different design spaces that the optimizer can traverse, while the rest of the optimization formulation is kept the same. We clarify the scope of this comparison in Sect. 5.2.

The identification of these aerostructural tradeoffs and quantification of the advantages of the "TwChTk" parametrization are some of the key findings of our work. Larger chords are associated with an increase in thrust and torque, but come at the expense potentially larger stresses and thus a mass penalty. The optimal solution thus results from a compromise that is tailored for the specific combination of operating conditions and objective formulation. Extensions of this study could leverage on local shape variables and a finer discretization of the blade in panels to reveal more about these tradeoffs, which could be further investigated by more detailed post-optimality analysis. We finally reiterate that more practical designs would be enabled by a multipoint or mixed-fidelity problem formulation to include constraints on stall margins and fatigue life, among others.

**References**

[1] Christian Bak, Frederik Zahle, Robert Bitsche, Taeseong Kim, Anders Yde, Lars Christian Henriksen, Anand Natarajan, and Morten Hansen. Description of the dtu 10 mw reference wind turbine. Technical Report DTU Wind Energy Report-I-0092, Danish Technical University, DTU Wind Energy Roskilde, DK, 2013.

[2] Denis-Gabriel Caprace, Adam Cardoza, Andrew Ning, Marco Mangano, Sicheng He, and Joaquim R. R. A. Martins. Incorporating high-fidelity aerostructural analyses in wind turbine rotor optimization. In *AIAA SciTech Forum*, January 2022.

[3] Mads H. Aa. Madsen, Frederik Zahle, Niels N. Sørensen, and Joaquim R. R. A. Martins. Multipoint high-fidelity CFD-based aerodynamic shape optimization of a 10 MW wind turbine. *Wind Energy Sciences*, 4:163–192, April 2019.